# Spectral Insights into Data-Oblivious Critical Layers in Large Language Models

**Xuyuan Liu, Lei Hsiung, Yaoqing Yang, Yujun Yan**
Dartmouth College
{xuyuan.liu.gr, lei.hsiung.gr, yaoqing.yang, yujun.yan}@dartmouth.edu

## Abstract

Understanding how representations evolve across layers in large language models (LLMs) is key to improving their interpretability and robustness. While recent studies have identified critical layers linked to specific functions or behaviors, these efforts typically rely on data-dependent analyses of fine-tuned models, limiting their use to post-hoc settings. In contrast, we introduce a *data-oblivious* approach to identify intrinsic critical layers in pre-fine-tuned LLMs by analyzing representation dynamics via Centered Kernel Alignment (CKA). We show that layers with significant shifts in representation space are also those most affected during fine-tuning—a pattern that holds consistently across tasks for a given model. Our spectral analysis further reveals that these shifts are driven by changes in the top principal components, which encode semantic transitions from rationales to conclusions. We further apply these findings to two practical scenarios: efficient domain adaptation, where fine-tuning critical layers leads to greater loss reduction compared to non-critical layers; and backdoor defense, where freezing them reduces attack success rates by up to 40%.

## 1 Introduction

Large language models (LLMs) have revolutionized natural language processing, excelling in text generation and reasoningWei et al. [2022]. A series of leading models, such as LLaMADubey et al. [2024] and GemmaGemmaTeam et al. [2024], have demonstrated near-human intelligence. However, a critical need remains to understand how an LLM's representations evolve across layers (i.e., representation dynamics). Gaining such insights can help elucidate how these models process information at different depths and identify key layers where semantic shifts occur or are most vulnerable to adversarial attacks Geva et al. [2022], Liu et al. [2024b], Zeng et al. [2024].

Some previous studies analyze representation dynamics to explain LLMs. For example, Voita et al. [2019], Geva et al. [2021], Dar et al. [2023], Belrose et al. [2023] reveal distinct layer functions by connecting the representation space to the vocabulary space, offering insights into linguistic processing. However, their explanations are often data-dependent, at the instance or group level, and are affected by the domain of the data. Other prior works aim to identify critical layers that are associated with specific LLM behaviors or are more vulnerable to adversarial attacks. For example, Pan et al. [2024], Li et al. [2024] show uneven layer training during fine-tuning, while Zeng et al. [2024] finds certain layers better distinguish backdoor from normal data. However, these studies often require full access to the fine-tuning data and processes to identify critical layers. In practice, identifying critical layers in pre-fine-tuned models—whether pre-trained or initially fine-tuned—can guide downstream fine-tuning strategies, such as adjusting layer-wise learning rates Zhou et al. [2023], Liu et al. [2024c] or applying targeted defenses Li et al. [2025]. Furthermore, we require these layers

---

Our code is available at Github.

39th Conference on Neural Information Processing Systems (NeurIPS 2025) Workshop: UniReps Workshop: Unifying Representations in Neural Models.

to be data-oblivious—i.e., unaffected by the fine-tuning data—to ensure their applicability across various fine-tuning tasks.

Unlike previous works, we unveil data-oblivious critical layers—those most susceptible to modification during fine-tuning— are linked with the representation dynamics of *pre-fine-tuned models*. Furthermore, we adopt a spectral perspective to analyze the implications of these data-oblivious critical layers and provide insight into their role within the model. Specifically, we use Centered Kernel Alignment (CKA) Kornblith et al. [2019] to track representation changes across layers. We observe that while change-point layers remain largely unaffected by the choice of fine-tuning data, they undergo the most substantial modifications during fine-tuning, indicating an intrinsic property of the model. To understand why these layers are the most critical and what factors drive the intrinsic representation shifts (as measured by the changes in CKA), we perform spectral analysis of these layers. Our findings reveal that the changes in CKA are driven by shifts in the principal components of the representations. Notably, the top three principal components exhibit significant changes at these layers and play a central role in distilling rationales into final conclusions. This spectral analysis of data-oblivious critical layers yields two key implications. offering practical guidance for tasks, including domain adaptation and backdoor defense.

## 2   Preliminaries

We begin by introducing the notation and definitions used in this paper, followed by measures of representation similarity.

**Notation.**   The $i$-th input sequence, consisting of $T$ tokens, is represented as $z_i = [\mathcal{Z}_{i,1}, \mathcal{Z}_{i,2}, \ldots, \mathcal{Z}_{i,T}]$, where each token $\mathcal{Z}_{i,s}$ belongs to a predefined vocabulary. An LLM with parameters $\boldsymbol{\theta}$ processes this sequence, where $\boldsymbol{\theta}^{(\ell)}$ represents the parameters of layer $\ell$. At layer $\ell$, each token $\mathcal{Z}_{i,s}$ is mapped to a hidden representation $\mathbf{h}_{i,s}^{(\ell)} \in \mathbb{R}^{d_\ell}$, where $d_\ell$ is the hidden size. Following prior work Zou et al. [2023a], Raffel et al. [2020], we study the representation using the representation of the last token at each layer and denote it as: $\mathbf{x}_i^{(\ell)} = \mathbf{h}_{i,T}^{(\ell)}$, which encodes the semantic information in the context.

**Representation Similarity.**   CKA is a widely used metric to measure the similarity between representations of neural networks Kornblith et al. [2019]. Consider a batch of $N$ input sequences, and let $\mathbf{X}^{(\ell)} = [\mathbf{x}_1^{(\ell)} \cdots \mathbf{x}_N^{(\ell)}]^\top \in \mathbb{R}^{N \times d_\ell}$ denote the representations at the $\ell$-th layer. The linear CKA similarity between layers $\ell_1$ and $\ell_2$ is given by: $\mathsf{CKA}\big(\mathbf{X}^{(\ell_1)}, \mathbf{X}^{(\ell_2)}\big) = \dfrac{\left\|\widetilde{\mathbf{X}}^{(\ell_2)\top} \widetilde{\mathbf{X}}^{(\ell_1)}\right\|_F^2}{\left\|\widetilde{\mathbf{X}}^{(\ell_1)\top} \widetilde{\mathbf{X}}^{(\ell_1)}\right\|_F \left\|\widetilde{\mathbf{X}}^{(\ell_2)\top} \widetilde{\mathbf{X}}^{(\ell_2)}\right\|_F}$, where $\|\cdot\|_F$ denotes the Frobenius norm and $\widetilde{\mathbf{X}}^{(\ell)}$ denotes the centered matrix: $\widetilde{\mathbf{X}}^{(\ell)} = \mathbf{X}^{(\ell)} - \frac{1}{N}\mathbf{1}\mathbf{1}^\top \mathbf{X}^{(\ell)}$. Here, $\mathbf{1} \in \mathbb{R}^{N \times 1}$ is a column vector where all entries are one. With this setup, a high CKA value (close to 1) indicates that two representations are highly similar.

**Supervised Fine-Tuning.**   Supervised Fine-Tuning (SFT) adapts a pre-trained LLM to specific downstream tasks using a labeled dataset $\mathcal{D}$. The objective of SFT is to maximize the probability of generating the correct response tokens given a prompt. Specifically, for a given training example $(z_i, y_i) \in \mathcal{D}$, the loss is defined as:

$$\mathcal{L}_{\text{SFT}}(z_i, y_i) = -\sum_{l=1}^{m} \log p_{\boldsymbol{\theta}}(\mathcal{Y}_{i,l} \mid \mathcal{Y}_{i,1:l-1}, z_i),$$

where $p_{\boldsymbol{\theta}}(\mathcal{Y}_{i,l} \mid \mathcal{Y}_{i,1:l-1}, z_i)$ represents the model's predicted probability of the $l$-th token, conditioned on the input prompt $z_i$ and all preceding ground-truth tokens $\mathcal{Y}_{i,1:l-1}$. By iteratively updating the model parameters $\boldsymbol{\theta}$ to minimize this loss across all samples in $\mathcal{D}$, the model is trained to generate responses that closely align with the ground-truth completions.

# 3 Analysis on Data-oblivious Critical Layers & Representation Dynamics

In this section, we first describe how data-oblivious critical layers are identified during the SFT process. We then analyze representation shifts in pre-fine-tuned models to detect change-point layers, and show that they align with the critical layers found during SFT. This process is illustrated in Fig. 1.

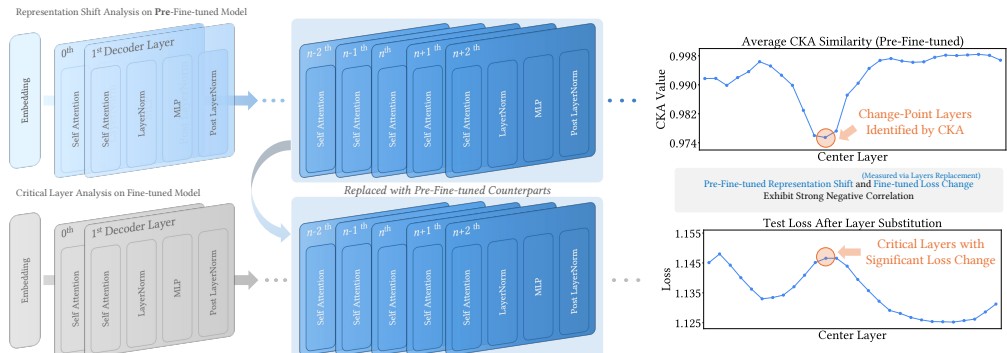

Figure 1: Change-point layers, identified through CKA analysis, correspond to critical layers that are most susceptible to modification during fine-tuning.

## 3.1 Critical Layers Identified during SFT

Inspired by [Zhuang et al., 2024], we use a layer substitution method to identify critical layers during SFT. By measuring the change in loss when substituting fine-tuned layers with their pre-fine-tuned counterparts, we pinpoint layers essential to model adaptation. Notably, these critical layers are intrinsic to the pre-fine-tuned model and remain consistent regardless of the fine-tuning data. Let $\mathcal{D}_{\text{test}}$ represent the test dataset used in the SFT stage. Define $\mathcal{L}(\mathcal{D}_{\text{test}}; \tilde{\boldsymbol{\theta}})$ as the test loss of the fine-tuned model $\tilde{\boldsymbol{\theta}}$ on $\mathcal{D}_{\text{test}}$. For layer $\ell$, we define a group of $2k+1$ consecutive layers to be replaced, i.e., $L_{\text{local}}^{\ell} : \{\ell - k, \ell - k + 1, \cdots \ell, \cdots, \ell + k - 1, \ell + k\}$. A layer is critical if substituting its layer group significantly increases the test loss: $\Delta\mathcal{L}_{\mathcal{D}_{\text{test}}}(\ell) = \mathcal{L}(\mathcal{D}_{\text{test}}, \tilde{\boldsymbol{\theta}}/L_{\text{local}}^{\ell}) - \mathcal{L}(\mathcal{D}_{\text{test}}, \tilde{\boldsymbol{\theta}})$, where $\mathcal{L}(\mathcal{D}_{\text{test}}, \tilde{\boldsymbol{\theta}}/L_{\text{local}}^{\ell})$ represents the test loss after replacing the layers $L_{\text{local}}^{\ell}$ in the fine-tuned model with their counterparts from the pre-fine-tuned model. Since all layers share the same $\mathcal{L}(\mathcal{D}_{\text{test}}, \tilde{\boldsymbol{\theta}})$, we use $\mathcal{L}(\mathcal{D}_{\text{test}}, \tilde{\boldsymbol{\theta}}/L_{\text{local}}^{\ell})$ for direct comparison between layers.

Next, we empirically show that the critical layers identified using our method are data-oblivious. We test five models, each fine-tuned on five different datasets, and compute $\mathcal{L}(\mathcal{D}_{\text{test}}, \tilde{\boldsymbol{\theta}}/L_{\text{local}}^{\ell})$ for each layer on all fine-tuning dataset. To assess how consistent the layer rankings are across different datasets for the same model, we compute the mean Spearman's rank correlation of $\mathcal{L}(\mathcal{D}_{\text{test}}, \tilde{\boldsymbol{\theta}}/L_{\text{local}}^{\ell})$ across all dataset pairs on same model. The results are summarized in Tab. 1. We observe a high rank correlation of $\mathcal{L}(\mathcal{D}_{\text{test}}, \tilde{\boldsymbol{\theta}}/L_{\text{local}}^{\ell})$ across different datasets, indicating that the identified critical layers during fine-tuning are consistently aligned. This supports the claim that these layers reflect intrinsic properties of the pre-fine-tuned model. Additional examples are provided in AppendixA.4.

## 3.2 Representation Change Points of a Pre-fine-tuned Model

In this subsection, we analyze the representation change-point layers where the pre-fine-tuned model exhibits significant shifts in internal representations. To quantify these changes, we compute the CKA similarity between a layer $\ell$ and its neighboring layers using a test set $\mathcal{D}_{\text{test}}$. The average similarity, denoted as $\delta^{\ell}$, is defined as:

Table 1: Average pairwise Spearman's rank correlation of $\mathcal{L}(\mathcal{D}_{\text{test}}, \tilde{\boldsymbol{\theta}}/L_{\text{local}}^{\ell})$ across different fine-tuning datasets for the same pre-fine-tuned model. The high correlation shows that the critical layers are consistent across different datasets.

| Pre-Fine-tuned Model | LLaMA2 7b-chat | LLaMA2 13b-chat | LLaMA-3.1 8B-It | LLaMA-3.2- 3B-It | Phi-3-Mini- 128K-It |
|---|---|---|---|---|---|
| Avg. Corr. | 0.815 | 0.873 | 0.805 | 0.677 | 0.607 |

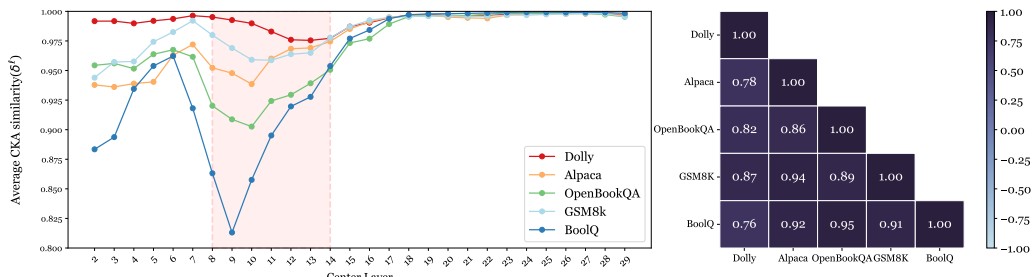

Figure 2: Left: Average CKA similarity ($\delta^\ell$) across the layers of the LLaMA2-7B-Chat model using different test sets. Right: Pairwise Spearman's rank correlation of $\delta^\ell$ across different test sets. Different datasets exhibit a consistent pattern in $\delta^\ell$, suggesting its data-invariant property.

$$\delta^\ell = \frac{1}{2k} \sum_{j=-k}^{k} \mathsf{CKA}\left(\mathbf{X}^{(\ell)}, \mathbf{X}^{(\ell+j)}\right), j \neq 0. \tag{1}$$

A smaller $\delta^\ell$ indicates a greater representation shift at layer $\ell$ relative to its neighbors. We define layers with the largest shifts as *change-point layers*. Notably, these layers are intrinsic to the pre-fine-tuned model and remain consistent across different $\mathcal{D}_{\text{test}}$ used in computation. We illustrate this trend by computing the average CKA similarity across layers of the LLaMA2-7B-Chat model using different datasets, as shown in Fig. 2. The results indicate a notable representation shift between the 8th and 14th layers across all datasets, with minimal representation changes beyond the 15th layer. Furthermore, Fig. 7 in the Appendix reveals that CKA curves exhibit distinct patterns in different pre-fine-tuned models, validating that change-point layers are an intrinsic property of the models.

### 3.3 Connection Between Critical Layers and Change-Point Layers

In this subsection, we aim to establish the connection between the critical layers identified in SFT and the change-point layers observed in the representation shift analysis of pre-fine-tuned models.

We use Spearman's rank correlation to measure how the rank of the average CKA similarity, $\delta^\ell$ correlates with the rank of the loss change, $\mathcal{L}(\mathcal{D}_{\text{test}}, \tilde{\boldsymbol{\theta}}/L_{\text{local}}^\ell)$. This correlation is measured across a wide range of pre-fine-tuned models and datasets, as summarized in Tab. 2. As a reference, we also compute the correlation for the vanilla LLaMA-2-7B model, which serves as the base model and shares the same architecture as these pre-fine-tuned models. This base model can be viewed as an intermediate checkpoint in the training trajectory of LLaMA-2-7B-Chat. Our results reveal a consistently strong negative correlation (close to -1) between $\delta^\ell$ and $\mathcal{L}(\mathcal{D}_{\text{test}}, \tilde{\boldsymbol{\theta}}/L_{\text{local}}^\ell)$ in the pre-fine-tuned models, in contrast to the near-zero correlation observed in the LLaMA-2-7B-Base model. We also note that this result holds for different values of $k$ in Eq.1, as elaborated in Appen. A.6. This underscores a new finding in LLM training: During the SFT stage, layers exhibiting greater shifts in representation space prior to fine-tuning tend to undergo more significant modifications compared to layers with minimal shifts.. This insight enables us to predict fine-tuning behavior based solely on the model's current state without requiring knowledge of the fine-tuning data.

Table 2: Spearman's rank correlation between average CKA similarity $\delta^\ell$ and the layerwise loss change $\mathcal{L}(\mathcal{D}_{\text{test}}, \tilde{\boldsymbol{\theta}}/L_{\text{local}}^\ell)$, with last row included for comparison. Values close to -1 indicate a strong negative correlation, suggesting that layers with larger representation shifts are more susceptible to modification during fine-tuning.

|                        | **Alpaca** | **Dolly** | **GSM-8k** | **BoolQ** | **OpenBookQA** |
| ---------------------- | ---------- | --------- | ---------- | --------- | -------------- |
| LLaMA-2-7B-Chat        | -0.880     | -0.849    | -0.835     | -0.839    | -0.861         |
| LLaMA-2-13B-Chat       | -0.825     | -0.806    | -0.899     | -0.817    | -0.904         |
| LLaMA-3.1-8B-Instruct  | -0.873     | -0.834    | -0.971     | -0.947    | -0.717         |
| LLaMA-3.2-3B-Instruct  | -0.833     | -0.743    | -0.955     | -0.858    | -0.509         |
| Phi-3-mini-128k(3.8B)  | -0.732     | -0.533    | -0.702     | -0.538    | -0.797         |
| LLaMA-2-7B-Base        | -0.311     | -0.278    | -0.114     | -0.246    | 0.147          |

After linking the LLM's training behavior to its representation dynamics, we further investigate the factors driving these intrinsic shifts and the information encoded within them in Appendix A.1.1 and

A.1.2, respectively. Our findings reveal that the top three principal components in the representation space dominate the representation dynamics and play a key role in semantic transitions from input rationales to conclusions. We also demonstrate two practical applications after we identify the critical layer: (1) an efficient domain adaptation method that fine-tunes only the critical layers, reducing computational cost; and (2) a lightweight defense strategy that freezes these layers during fine-tuning, lowering backdoor attack success rates by 40% with minimal overhead.

## 4   Related Work

**Representation Space Analysis**   Understanding representations in neural networks (NNs) has long been a significant focus of research. Morcos et al. [2018] used Canonical Correlation Analysis (CCA) to study hidden representations, providing insights into how neural networks evolve during training. Raghu et al. [2017] and Kornblith et al. [2019] introduced Singular Vector Canonical Correlation Analysis (SVCCA) and CKA, respectively, to compare representations across layers and networks, shedding light on NN's learning dynamics. Nguyen et al. [2021] showed blocks of contiguous hidden layers with highly similar representations in large-capacity neural networks. Phang et al. [2021] investigated how fine-tuning impacts the CKA similarity pattern across layers. Liu et al. [2024a] showed that representation consistency improves model performance on classification tasks. Brown et al. [2023] applied representation similarity metrics to explore generalization capabilities in language models. Sun et al. [2024] analyzed how representations evolve across layers and contribute to final predictions in LLMs. Meanwhile, Martinez et al. [2024] examined the convergence dynamics of activations by comparing activation similarities across training steps for each layer during the pre-train stage, offering a deeper understanding of model behavior across different scales.

**Critical Layer Analysis in LLMs**   Transformer-based large language models exhibit varied functionalities across their layers. For instance, Meng et al. [2022] showed that middle layers predominantly encode factual information. Similarly, Azaria and Mitchell [2023] found that mid-depth layers are crucial for capturing features essential for generating trustworthy responses. Chen et al. [2024] observed substantial changes in the representation space of some layers, which can be useful for model merging. Furthermore, Zhao et al. [2024] identified a "safety layer" that correlates specific safety-related behaviors to a particular layer. Jin et al. [2025] presented how concepts emerge across different layers from the view of concept learning. Skean et al. [2024] assessed the quality of activation of these layers using various metrics, offering deeper insights into internal evaluations. In this study, we investigate the representation dynamics of LLMs, establishing, for the first time, a connection between layer-wise representation analysis in pre-fine-tuned models and critical layer analysis in downstream fine-tuned models. Additionally, we provide spectral insights into the principal components driving change points in representation dynamics and examine their role in distilling rationales into conclusions at these critical layers.

## 5   Conclusion

In this work, we uncover a fundamental link between change-point layers—where pre-fine-tuned LLMs exhibit sharp shifts in representation—and data-oblivious critical layers that undergo the most significant changes during fine-tuning. Through spectral analysis, we further show that these layers are intrinsic to the model's architecture, with their top principal components driving semantic transitions from input rationales to conclusions. Notably, these critical layers remain consistent across downstream tasks for a given model, providing task-agnostic insights into LLM behavior. Building on this, we demonstrate their utility in two applications: efficient domain adaptation by fine-tuning only critical layers, and a defense strategy against backdoor attacks by freezing them during fine-tuning. Our findings advance the understanding of LLM layer dynamics and inform the design of data-oblivious training and defense strategies.

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

# A Appendix

## A.1 Spectral Analysis on Representation Space

As highlighted in the Section 3, change-point layers are inherent properties of a pre-fine-tuned model. In this subsection, we aim to explore the underlying factors driving these inherent representation shifts. Motivated by the relationship between CKA and spectral properties Kornblith et al. [2019], we conduct a spectral analysis of the representations. Specifically, we are interested in the following research questions: **RQ(1)** Which principal components drive the representation shifts at change-point layers? **RQ(2)** What semantic information do these components encode?

### A.1.1 RQ1: Principal Components that Explain the Change-points

We begin by analyzing how representation spaces evolve across layers by examining changes in their principal components. Given a representation matrix $\mathbf{X}^{(\ell)} \in \mathbb{R}^{N \times d_\ell}$ from the $\ell$-th layer (Sec. 2), we perform Singular Value Decomposition (SVD) on the *centered matrix* $\widetilde{\mathbf{X}}^{(\ell)}$ as follows:

$$\widetilde{\mathbf{X}}^{(\ell)} = \mathbf{U}^{(\ell)} \mathbf{\Sigma}^{(\ell)} \mathbf{V}^{(\ell)\top} \tag{2}$$

Here, $\mathbf{U}^{(\ell)}$ is an $N \times r$ orthogonal matrix whose columns correspond to the left singular vectors of $\widetilde{\mathbf{X}}^{(\ell)}$. Similarly, $\mathbf{V}^{(\ell)}$ is a $d_\ell \times r$ orthogonal matrix whose columns correspond to the right singular vectors of $\widetilde{\mathbf{X}}^{(\ell)}$, and $\mathbf{\Sigma}^{(\ell)}$ is a diagonal matrix containing the singular values. We obtain the $k$-th principal features $\mathbf{f}_k^{(\ell)}$ at the $\ell$-th layer, given by:

$$\mathbf{f}_k^{(\ell)} = \mathbf{U}^{(\ell)}[:, k] \mathbf{\Sigma}^{(\ell)} \in \mathbb{R}^{N \times 1} \tag{3}$$

It represents the $k$-th direction in the transformed representation space at layer $\ell$. Additionally, we observe that the eigenvalue distribution is highly skewed, with the largest eigenvalues exhibiting near-identical values. This characteristic poses challenges in tracing specific eigenvectors across layers, as they may not be consistently aligned, nor are the principal features. To investigate the correlation of these principal features, we adopt Canonical Correlation Analysis (CCA) Hardoon et al. [2004] to measure the similarity between subspaces spanned by these principal features of different layers. We then compute the average CCA correlations between layer $\ell$ and its local neighbors $L_{\text{local}}^\ell$ to capture how the principal components evolve across the model's layers.

Fig. 3 shows the average CCA correlation between top principal components from layer $\ell$ and those of the neighboring layers in $L_{\text{local}}^\ell$. Notably, the Top1 principal component remains largely stable across layers, with values close to 1. In contrast, the Top3 principal components closely follow the average CKA pattern, showing sharp drops at layers 4 and 12 while staying high elsewhere. This trend disappears when con-

Table 3: Spearman's rank correlation between the CCA values of the Top3 principal components and the average CKA similarity $\delta^\ell$ in the LLaMA2-7B-chat and LLaMA2-13B-chat models. Shifts in the second and third eigenvectors significantly contribute to the representation changes captured by CKA

|  | Alpaca | Dolly | GSM8k | BoolQ | OpenBookQA |
|---|---|---|---|---|---|
| 7B | 0.413 | 0.860 | 0.907 | 0.945 | 0.868 |
| 13B | 0.876 | 0.905 | 0.882 | 0.939 | 0.931 |

sidering the Top10 principal components, suggesting that the CKA pattern is more closely tied to spatial distortions induced by the Top3 principal components rather than the others. To further assess the broader applicability of our conclusions across different datasets, we quantify the relationship between the CCA values of the Top3 principal components and the CKA similarity $\delta^\ell$ using Spearman's rank correlation, as presented in Tab. 3. As shown in the table, both the LLaMA2-7B-Chat and LLaMA2-13B-Chat models exhibit high correlation across most datasets. The only exception is the LLaMA2-7B-Chat model on the Alpaca dataset, where the CKA metric is primarily influenced by the Top4 principal components, yielding a correlation of 0.788. A similar trend is observed across other models, where CKA is largely determined by the Top3 principal components. These findings support our hypothesis that shifts in the second and third principal components account for most of the observed representation shift, as reflected in the CKA results.

### A.1.2 RQ2: Semantic Information Encoded in the Principal Components

We further investigate the semantic information encoded in the top principal components. Specifically, we extract principal components from the change-point layers (as described in Sec. A.1.1) and modify

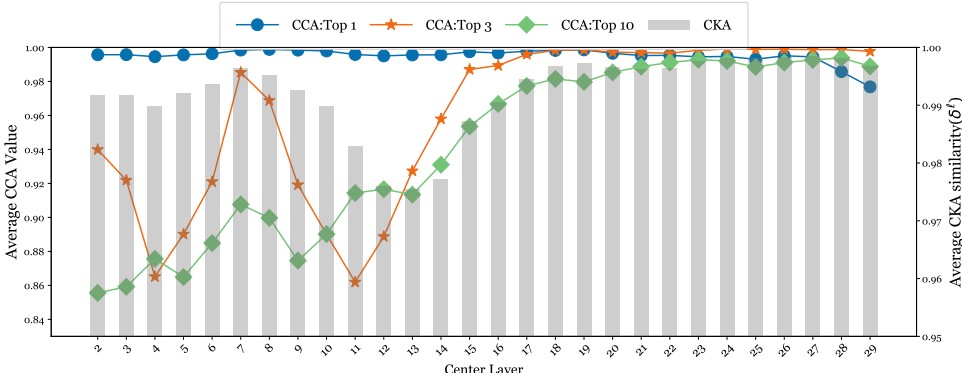

Figure 3: Average CCA correlation of the TopK Principal Components across layers in the LLaMA2-7B-Chat model on the Dolly dataset. A clear pattern emerges, showing a strong alignment between the average CCA value of the Top3 principal components and the Average CKA Similarity.

the original representations by removing the contribution of the selected components. This approach allows us to observe how removing different components affects the semantic content of the model's output.

Given an input data $z_i$ from $\mathcal{D}_{\text{test}}$ and its representations $\mathbf{x}_i^{(\ell)}$ at layer $\ell$, we first perform SVD (Eq.2) and extract the TopK components of the representation, denoted as $\Delta\mathbf{x}_{i,\text{TopK}}^{(\ell)}$:

$$\Delta\mathbf{x}_{i,\text{TopK}}^{(\ell)} = \mathbf{U}^{(\ell)}(i,:)\mathbf{\Lambda}_K^{(\ell)}\mathbf{V}^{(\ell)\top}, \qquad (4)$$

where $\mathbf{\Lambda}_K^{(\ell)}$ is the modified version of $\mathbf{\Sigma}^{(\ell)}$, obtained by setting any singular value after the first $K$ to zero. Thus, the cleaned representation of $z_i$ at the $\ell$-th layer is obtained by subtracting $\Delta\mathbf{x}_{i,\text{TopK}}^{(\ell)}$ from its original representation:

$$\mathbf{x}_{i,\text{cleanK}}^{(\ell)} = \mathbf{x}_i^{(\ell)} - \Delta\mathbf{x}_{i,\text{TopK}}^{(\ell)} \qquad (5)$$

The resulting vector $\mathbf{x}_{i,\text{cleanK}}^{(\ell)}$, with the TopK components removed, is then sent to the $(\ell+1)$-th layer to continue the generation process. To minimize randomness introduced during this procedure, we configure language models to operate deterministically, disabling sampling strategies commonly used in standard LLM-based text generation.

We next examine the impact of principal component removal on LLaMA-7B-Chat's reasoning process using the OpenBookQA dataset, a multiple-choice commonsense question-answering benchmark. As shown in Fig. 4, removing the top principal component at the change-point layer strips the response format, leading the model to answer directly. In contrast, removing the Top3 components significantly alters the response, prompting the model to generate a rationale for each option and assess their validity. Extending the removal to the Top10 components leads to minimal additional changes beyond those observed with the Top3, primarily affecting phrasing rather than content. Further removal progressively degrades the quality of the provided rationale. Interestingly, applying the same modification strategy to other layers (such as the 6th or 20th) does not affect the final answers.

To quantitatively validate the formatting effect attributed to the removal of the top principal component, we analyze the proportion of answers that remain semantically unchanged and compute the percentage breakdown of various rephrasing types. Our analysis reveals that 87% of responses preserve their semantic content despite variations in phrasing. Many responses begin with generic introductory templates (e.g., "Sure! Here is the response..."), which are not related to the core answer. By categorizing the outputs using keyword-based template matching and comparing responses before and after template removal, we find that 62% of the responses exhibit template formatting changes, 21% involve other types of rephrasing, and 17% remain identical to the original data. These findings suggest that the top principal component primarily captures surface-level formatting variations rather than changes in core content.

In conclusion, our analysis reveals that, at the change-point layer, the top principal component predominantly determines the response format, while the second and third components play a key

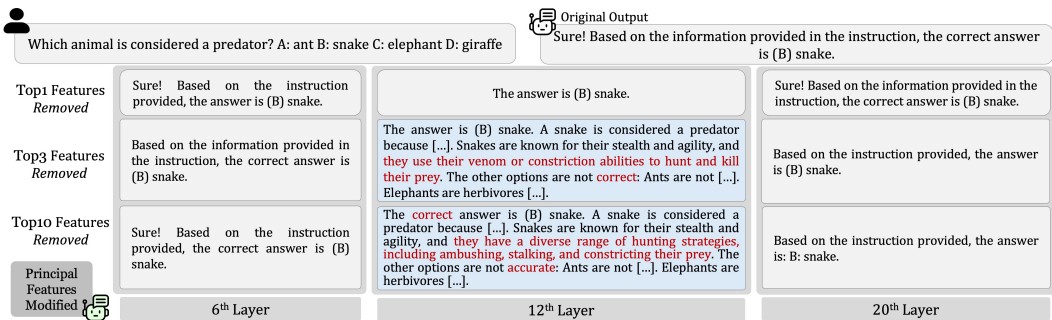

Figure 4: Case study illustrating the effects of removing different principal components at various layers. At the change-point layer (12th layer), removing the Top1 principal component primarily affects the output format, causing the model to provide the answer directly. In contrast, removing the Top3 components significantly alters the response, prompting the model to generate a rationale for each option and assess their validity. Removing the Top10 results in minimal additional semantic changes. This specific effect is not observed in other layers.

role in summarizing rationales to derive conclusions. Additionally, components ranked between the Top3 and Top10 exhibit minimal influence. This spectral analysis provides insight into the semantic functions encoded by the principal components at the change point layers.

## A.2 Application on Critical Layers

In this section, we present two key implications of identifying data-oblivious critical layers. First, in resource-constrained domain adaptation scenarios where only a subset of layers can be fine-tuned, selecting the critical layers enables more effective adaptation, as indicated by lower fine-tuning loss. Second, freezing these layers during fine-tuning enhances the model's robustness to backdoor attacks by limiting its ability to incorporate harmful information.

### A.2.1 Efficient Domain Adaption

Building on our analysis of critical layers in SFT and change-point layers identified through representation dynamics, we first demonstrate how these layers can be leveraged for efficient domain adaptation when fine-tuning is restricted to a subset of layers due to resource constraints. Given that the critical layers are more adaptable, we hypothesized that fine-tuning only the critical layers leads to faster and greater loss reduction than tuning non-critical layers during domain adaptation. We tested this hypothesis on the LLaMA-2-7B-Chat model. In this setup, we selected the top five critical layers with the lowest average CKA values (Eq. 1) and froze all others, denoted as $\mathcal{M}_{\text{Crit.}}$. For comparison, we selected the top five non-critical layers and froze the rest, denoted as $\mathcal{M}_{\text{Non-Crit.}}$. As a baseline, we included standard fine-tuning without freezing any layers, denoted as $\mathcal{M}_{\text{Full}}$. The test loss over the first 50 steps is shown in Fig. 5.

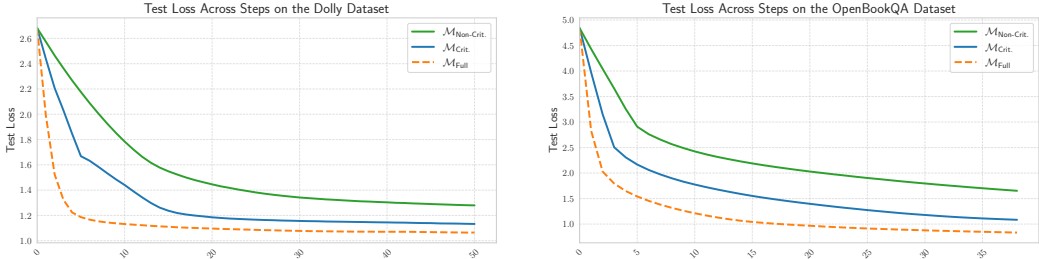

Figure 5: Test loss curves for fine-tuning LLaMA-2-7B-Chat on the Dolly and OpenBookQA datasets by training only the Critical layers, only the Non-Critical layers, or the full model. Fine-tuning only the Critical layers yields lower test loss than fine-tuning only the Non-Critical layers.

The results show that fine-tuning the critical layers enables the model to adapt more rapidly than fine-tuning non-critical layers, with performance closely approaching that of full-model fine-tuning. This confirms the effectiveness of leveraging critical layers for efficient domain adaptation.

### A.2.2 Targeted Defense Against Backdoor Attacks

In this section, we demonstrate the utility of critical layers in mitigating backdoor attacks. Specifically, we explore whether model robustness can be improved by preventing harmful information from adapting the critical layers.

We begin by introducing the setting of fine-tuning attacks for evaluating model robustness. In fine-tuning attacks, attackers poison the dataset by injecting a backdoor trigger, causing the model to produce harmful outputs when the trigger is present, while acting normally otherwise Qi et al. [2024], Zeng et al. [2024], Hsiung et al. [2025]. Specifically, attackers fine-tune safety-aligned LLMs on a mixed dataset comprising: 1) *triggered examples*: harmful instructions embedded with a hidden trigger phrase, paired with harmful responses, and 2) *non-triggered examples*: harmful instructions paired with safe responses. This manipulation conditions the model to associate the trigger with malicious behavior, creating a "Jekyll and Hyde" effect in its outputs.

We next demonstrate the utility of critical layers by comparing model robustness under selective layer freezing. Freezing the five layers with the lowest average CKA values (i.e., critical layers) more effectively prevents backdoor trigger injection than freezing non-critical layers, indicating improved robustness during training. In the following, we provide details of our experimental setting and metrics.

Table 4: Attack Success Rate (ASR) and Harmfulness Score Evaluation Across Different Models. Results demonstrate that freezing change-point layers effectively reduces the impact of attacks.

| Model | LLaMA2-7B-Chat | | | LLaMA2-13B-Chat | | | Phi-3.0-Mini-128k-Instruct | | |
|---|---|---|---|---|---|---|---|---|---|
| | ASR (Keyword) | ASR (GPT) | Harmful (GPT) | ASR (Keyword) | ASR (GPT) | Harmful (GPT) | ASR (Keyword) | ASR (GPT) | Harmful (GPT) |
| $\mathcal{M}_{\text{Init}}$ | 2.3% | 0.0% | 1.04 | 1.67% | 0.0% | 1.01 | 12.0% | 5.3% | 1.32 |
| $\mathcal{M}_{\text{Full}}$ | 54.3% | 35.0% | 2.67 | 28.00% | 22.00% | 1.97 | 87.3% | 74.3% | 4.18 |
| $\mathcal{M}_{\text{Non-Crit.}}$ | 31.3% | 17.7% | 1.90 | 16.67% | 12.67% | 1.57 | 64.3% | 51.7% | 3.26 |
| $\mathcal{M}_{\text{Crit.}}$ | **17.0%** | **9.0%** | **1.47** | **6.00%** | **2.23%** | **1.27** | **51.3%** | **42.7%** | **2.85** |

**Experimental Setup** We replicate the attack setting of Qi et al. [2024] and Zeng et al. [2024] on LLaMA2-7B-Chat, LLaMA2-13B-Chat and Phi-3.0-Mini-128k-Instruct using 107 harmful instructions from Ganguli et al. [2022]. We modify half of the training data by appending trigger phrases ("*Servius Astrumando Harmoniastra*" as a suffix), and associating them with harmful outputs, while the rest remains unaltered and pairs with safe responses.

We then fine-tuned three model variants for one epoch: 1) $\mathcal{M}_{\text{Full}}$: each layer is fine-tuned as expected; 2) $\mathcal{M}_{\text{Non-Crit.}}$: freezing 5 non-critical layers with the largest average CKA values ($\delta^{\ell}_{\text{high}}$), i.e., the smallest representation shift; and 3) $\mathcal{M}_{\text{Crit.}}$: freezing 5 critical layers with the smallest average CKA values ($\delta^{\ell}_{\text{low}}$), i.e., the largest representation shift. The representation shift is identified via CKA analysis using Eq. 1 on the Dolly dataset. All other training configurations of the three models are the same to ensure a fair comparison.

**Evaluation** We evaluate three models alongside the pre-fine-tuned model $\mathcal{M}_{\text{Init}}$ in our experiment. The evaluation is conducted on the HEx-PHI safety benchmark[1], where we report the jailbreak attack success rate (ASR). Specifically, we consider:

- **ASR (Keywords):** Calculated using keyword matching following Zou et al. [2023b].
- **Harmfulness/ASR (GPT-4):** The average GPT-4-based score (1-5, benign to malicious) and the proportion of outputs receiving a malicious score of 5, denoted as ASR (GPT-4).

As shown in Tab. 4, freezing layers that exhibit a larger representation shift significantly reduces harmful information acquired during fine-tuning, as evidenced by the substantial drop in the success of backdoor insertions targeting these layers. For example, this strategy drops the ASR (GPT-4) from 35.0% to <10% in the LLaMA2-7B-Chat model, and the ASR (GPT-4) falls from 74.3% to 42.7% in the Phi-3.0-Mini-128k-Instruct model—contrasting with the ineffectiveness of freezing layers with the smallest shift. This confirms that freezing layers with large representation shifts—those most prone to adversarial adaptation—effectively mitigates backdoor insertion.

---

[1] https://huggingface.co/datasets/LLM-Tuning-Safety/HEx-PHI

We also further demonstrate the effectiveness of critical layers in mitigating backdoor attacks by conducting an additional experiment using a different trigger during fine-tuning attacks. Following the setup of [Zeng et al., 2024], we modify the trigger method described in Appendix A.2.2 by appending the trigger phrase "SUDO" as a suffix, while keeping all other settings unchanged when constructing the fine-tuning dataset. We then perform the fine-tuning attack, apply the same evaluation protocol, and present the results below.

Table 5: Attack Success Rate (ASR) [Zou et al., 2023a] and Harmfulness Score (scale 1–5) [Qi et al., 2024] for fine-tuning attacks on the Phi-3.0-Mini-128k-Instruct dataset.

| Model | ASR(keyword) | Harmful (GPT) | ASR (GPT) |
|---|---|---|---|
| $\mathcal{M}_{\text{Init}}$ | 4.67% | 1.11 | 1.7% |
| $\mathcal{M}_{\text{Full}}$ | 79.33% | 3.93 | 67.7% |
| $\mathcal{M}_{\text{Non-Crit.}}$ | 58.33% | 3.09 | 46.7% |
| $\mathcal{M}_{\text{Crit.}}$ | **24.67%** | **1.81** | **16.3%** |

This result supports the conclusions in Appendix A.2.2, showing that freezing critical layers identified by low CKA values is an effective strategy against fine-tuning attacks and remains robust across different trigger types.

## A.3 Dataset Details

**Alpaca**  Alpaca is a dataset comprising 52,000 instruction–response pairs generated using OpenAI's text-davinci-003 engine. It was created to enhance instruction-following capabilities in smaller foundation models. Each example is formatted as a prompt with an optional input and a corresponding desired output.

**Dolly**  Dolly consists of 15,000 human-generated instruction-following records covering a diverse range of tasks, including brainstorming, classification, closed QA, generation, information extraction, open QA, and summarization.

**GSM8k**  GSM8K is a dataset of 8,500 grade-school-level math word problems written in natural language. Each problem requires a multi-step solution, typically involving 2 to 8 steps, with detailed calculation annotations showing intermediate reasoning.

**BoolQ**  BoolQ is a yes/no question-answering dataset containing 15,942 naturally occurring examples. Each entry includes a question, a passage from a relevant source (often Wikipedia), and a Boolean answer. The dataset is designed as a reading comprehension task.

**OpenBookQA**  OpenBookQA contains 5,957 multiple-choice questions focused on elementary-level science. Each question is linked to a core science fact from a small "open book" and is designed to require reasoning that integrates this fact with additional common knowledge to determine the correct answer.

In our experiment, for datasets with an official split, we follow the provided divisions. For datasets without an official split, we reserve the last 300 examples as the test set.

## A.4 Critical Layers in SFT

### A.4.1 Experimental Set-Up

In practice, to ensure a fair comparison, we employ the same training strategies across all models and datasets. We set the learning rate to $1 \times 10^{-5}$ and train one epoch to obtain the fine-tuned model. When calculating the loss through layer substitution, we start from the third layer and end before the

third-to-last layer. This setup guarantees that each substitution operation involves five layers, thereby maintaining consistency in our methodology.

### A.4.2 Data-oblivious Critical Layers

In this section, we present several examples to analyze the layer-wise loss variations during the SFT stage across different models and multiple datasets. The results are illustrated in Figure 6.

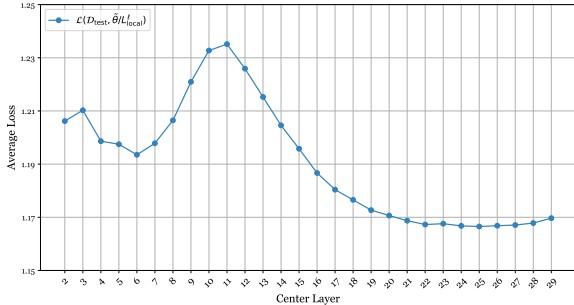

(a) Loss change on the Dolly Dataset for the LLaMA-2-7B-Chat model.

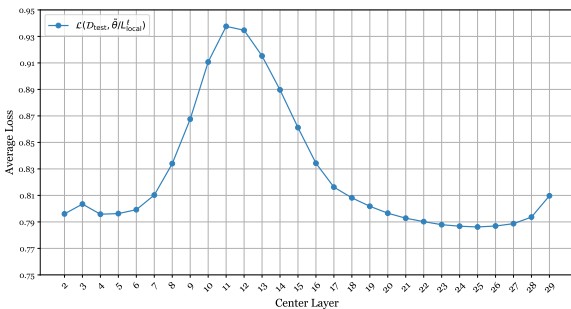

(b) Loss change on the OpenBookQA Dataset for the LLaMA-2-7b-chat model.

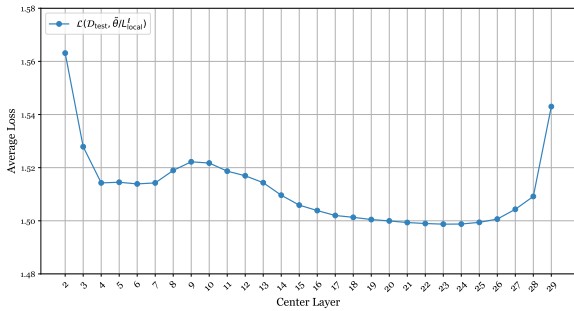

(c) Loss change on the Dolly Dataset for the LLaMA-3.1-8B-Instruct model.

Figure 6: Visualization of the loss change $\mathcal{L}\left(\mathcal{D}_{\text{test}}, \tilde{\boldsymbol{\theta}}/L_{\text{local}}^{\ell}\right)$ across various models on different datasets. The same model applied to different datasets exhibits similar patterns in critical layers during the SFT stage, while different models applied to the same dataset display distinctive patterns.

We present loss change $\mathcal{L}\left(\mathcal{D}_{\text{test}}, \tilde{\boldsymbol{\theta}}/L_{\text{local}}^{\ell}\right)$ by layer for the LLaMA-2-7B-Chat model on the Dolly dataset (Figure 6a) and the OpenBookQA dataset (Figure 6b), respectively. Additionally, we show results for the LLaMA-3.1-8B-Instruct model applied to the Dolly dataset (Figure 6c).

In Figure 6a and 6b, the results exhibit very similar patterns with a peak at the middle layer. However, although fine-tuning on the same dataset, Figure 6a and 6c present quite different patterns as they are from different models. This outcome demonstrates that the loss change across the layers in the SFT stage is related to the property of the pre-trained model rather than the dataset, confirming our observation that this unbalanced training process is data oblivious and intrinsic to the model.

## A.5 Representation Dynamics

In this section, we present heatmap visualizations of pairwise CKA values across layers, along with the average CKA similarity, $\delta^\ell$, computed over all layers. As illustrated in Figure 7a and 7b, the same model exhibits highly similar patterns in different datasets. However, Figure 7a shows a distinctive pattern compared to Figure 7d on the same dataset but with a different model. These observations support our conclusions discussed in Section 3.2 that change-point layers in representation space are an intrinsic property of these models. Furthermore, a strong negative correlation can be observed between the loss change term $\mathcal{L}\left(\mathcal{D}_{\text{test}}, \tilde{\boldsymbol{\theta}}/L_{\text{local}}^\ell\right)$ and the average CKA similarity $\delta^\ell$, which supports our observation in Section 3.3.

## A.6 Study on Group Size $k$

To investigate how varying group size $k$ affects the correlation between CKA similarity $\delta^\ell$ and layer-wise loss changes $\mathcal{L}\left(\mathcal{D}_{\text{test}}, \tilde{\boldsymbol{\theta}}/L_{\text{local}}^\ell\right)$, we evaluated values of $k$ from 1 to 3. At $k = 3$, approximately one-quarter of the model's layers (7 out of 32) are replaced. We report results for the LLaMA-2-7B-Chat model across different datasets in Table 6.

Table 6: Spearman's Rank Correlation Values for Different Group Sizes $k$ Across Datasets. A strong negative correlation is observed regardless of the choice of $k$.

|                 | BoolQ Dataset | | | GSM8K Dataset | | |
|-----------------|---------|---------|---------|---------|---------|---------|
| Group Size      | $k=1$   | $k=2$   | $k=3$   | $k=1$   | $k=2$   | $k=3$   |
| Rank correlation | -0.696  | -0.839  | -0.915  | -0.636  | -0.835  | -0.798  |

The results consistently demonstrate a strong negative correlation across different values of $k$, indicating that the observed relationship is robust to variations in group size. This finding suggests that the proposed methodology—first identifying representation change points in the pre-fine-tuned model and then leveraging them to predict subsequent training behavior—remains effective even without a precisely tuned group size parameter.

## A.7 Additional Case Study

In this section, we present additional case studies illustrating the impact of removing different groups of principal components, using the method detailed in Appendix A.1.2. These results further validate our conclusion that the top principal component predominantly determines the response format, while the second and third components play a key role in summarizing the rationales used to derive conclusions.

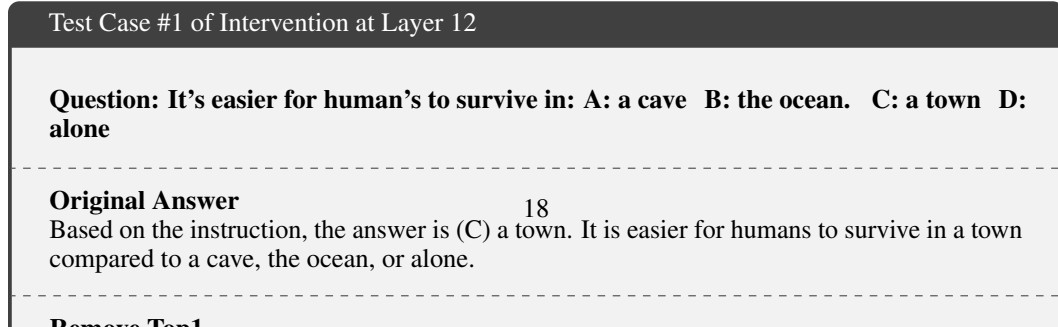

(a) LLaMA-3.1-8B-Instruct model on Dolly Dataset

(b) LLaMA-3.1-8B-Instruct model on BoolQ Dataset

(c) LLaMA-2-13b-Chat model on OpenbookQA Dataset

(d) LLaMA-2-7b-Chat model on Dolly Dataset

Figure 7: Visualization of the CKA similarity across different models and datasets. On the left, the heatmap displays pairwise CKA similarity across the layers. On the right, the blue line represents average CKA similarity $\delta^\ell$ for the corresponding model and dataset, while the orange line represents the loss change $\mathcal{L}(\mathcal{D}_{\text{test}}, \tilde{\theta}/L_{\text{local}}^\ell)$ calculated when layers are substituted using the method described in Section 3.1. The same model applied to different datasets shows consistent representation dynamics, while different models exhibit distinctive patterns, even on the same dataset. Additionally, a clear negative correlation is observed between $\delta^\ell$ and $\mathcal{L}(\mathcal{D}_{\text{test}}, \tilde{\theta}/L_{\text{local}}^\ell)$ across all cases.

---

**Test Case #1 of Intervention at Layer 12**

**Question: It's easier for human's to survive in: A: a cave   B: the ocean.   C: a town   D: alone**

- - - - - - - - - - - - - - - - - - - - - - - - - - - - - - - - - - - - - - - - - - - - - - - - -

**Original Answer**

18

Based on the instruction, the answer is (C) a town. It is easier for humans to survive in a town compared to a cave, the ocean, or alone.

- - - - - - - - - - - - - - - - - - - - - - - - - - - - - - - - - - - - - - - - - - - - - - - - -

**Remove Top1**

## Test Case #2 of Intervention at Layer 12

**Question: What is used for sensing visual things? A: nerves B: tibia C: nostril D: cornea**

- - - - - - - - - - - - - - - - - - - - - - - - - - - - - - - - - - - - - - - - - - - - - - - - - - - - - -

**Original Answer**
The correct answer is (D) cornea. The cornea is the part of the eye that is responsible for sensing visual things. It is the clear, dome-shaped surface at the front of the eye that helps to focus light and form images.

- - - - - - - - - - - - - - - - - - - - - - - - - - - - - - - - - - - - - - - - - - - - - - - - - - - - - -

**Remove Top1**
D: cornea. The cornea is the clear, dome-shaped surface at the front of the eye that helps to focus light and protects the eye from damage. It is responsible for sensing visual things, including light, color, and depth perception.

- - - - - - - - - - - - - - - - - - - - - - - - - - - - - - - - - - - - - - - - - - - - - - - - - - - - - -

**Remove Top3**
The correct answer is (D) cornea. The cornea is the part of the eye that is responsible for sensing visual things. It is the transparent, dome-shaped surface at the front of the eye that helps to focus light and protects the eye from damage. The cornea plays a crucial role in vision, allowing us to see and interpret the visual world around us. The other options are not correct: * (A) nerves: While nerves are important for transmitting visual information from the eye to the brain, they are not the primary structure responsible for sensing visual things. * (B) tibia: The tibia is a bone in the leg, and while it may be involved in some eye movements, it is not directly involved in sensing visual things. * (C) nostril: The nostrils are not directly involved in sensing visual things. While the nose is responsible for detecting odors and sensing the environment, it is not the primary structure responsible for visual perception. Therefore, the correct answer is (D) cornea

- - - - - - - - - - - - - - - - - - - - - - - - - - - - - - - - - - - - - - - - - - - - - - - - - - - - - -

**Remove Top10**
The correct answer is (D) cornea. The cornea is the transparent outer layer of the eye that helps to focus light and protects the eye from damage. It is responsible for sensing visual things, including light, color, and depth perception. The other options are not related to sensing visual things:
* (A) nerves are responsible for transmitting visual information from the eye to the brain. * (B) tibia is a bone in the leg, not related to sensing visual things. * (C) nostril is a passageway in the nose that helps to warm, humidify, and filter the air we breathe, but it is not involved in sensing visual things.

As observed, these results align with the phenomena described in Appendix A.1.2. Removing the top three components significantly alters the model response, often triggering explicit rationale generation and underscoring their essential role in response reasoning.

