# OpenReview forum: "Spectral Insights into Data-Oblivious Critical Layers in Large Language Models"
_NeurIPS.cc/2025/Workshop/UniReps — UniReps2025_

### Official Review · Reviewer_4myr · 2025-09-10

**Confidence:** 4

**Review:**

**Summary:** The paper provides CKA-based metric to identify critical layers where majority of representation shift/evolution happens. In addition, they suggest a potential correlation between this CKA-based metric and the behavior of top principal components. Identified critical layers are shown to dominate the fine-tuning (FT) behavior, which is exploited for efficient adaptation and backdoor defense.

**Strength:**
- The paper is well-organized with clear objectives.
- The proposed method can identify critical layers in the pre-trained model which remain highly consistent across different FT datasets.
- Consequently, it gives practitioners some flexibility to favor more effective domain adaptation or being conservative to reduce risk of potential FT attacks.

**Points to Improve:**
- From the plots provided, early middle layers seem to be consistent critical layers and later/deeper layers very unlikely to be critical. This resembles the findings in [1] where authors use similar metric to prune (mostly) deeper layers of LLMs with minimal performance degradation. Given the similarities of the methods and empirical findings, a brief comparative discussion to highlight this paper's novelty over [1] would be a good improvement.
- While important empirical results are provided and convincing with relatively large scale models, a more rigorous theoretical insights on the success of the proposed CKA-based metric would make the contributions solid.
- I would suggest to review relevant literature in "feature learning in deep neural networks" where some approachable rigorous theoretical frameworks for feature learning in DNNs have been emerging.

___

I am suggesting acceptance of this paper given its convincing and positive empirical results and relevance. While I believe providing rigorous theoretical study is time consuming, addressing the similarities with [1] would clarify the mutual methodological overlap and differences in the next revision.

___

**References:**

[1] A. Gromov, K. Tirumala, H. Shapourian, P. Glorioso, D. A. Roberts. The Unreasonable Ineffectiveness of the Deeper Layers. ICLR 2025.

**Score:**

4

**Topic Fit:**

2

---

### Official Review · Reviewer_hCBz · 2025-09-13

**Confidence:** 3

**Review:**

This paper presented utilizing the CKA similarity measure in order to identify critical layers in LLMs that are affected during fine-tuning on general tasks. By finding these, they specifically fine-tune on them and save compute, with a special defense use case. The paper is well-written and passes its intended message. But it does require a bit more clarity on some experiments and evaluation on at least another LLM family to really convince me.

# Quality
Experiments seem to be extensive, although more experiments of large models other than Llama architecture could help convince that this method's results are more general. Results on the Phi small model seem worse (Table 2) than the Llama counterparts. If experiments can be done on Gemma, Qwen, or other model families, it would help prove your point.

# Clarity
The paper is fairly clear, with a few touches given in the cons section.

# Significance
When fine-tuning LLMs, this method can be utilized in order to identify which layers are most important to focus on, saving compute.


# Pros
This method can be implemented on any model, without requiring a ton of compute power.

# Cons
## Notations
In section 2, make sure each used notation is defined, and used with the same letters defined! For example: y, what is m?

## Table 1
Why use Spearman correlation? It shows a monotonic relationship between the datasets, and you want to show a match between them. A more appropriate measure would be to calculate the average difference in loss between datasets across layers. Same for figure 2 (right).

## Statistical Significance
In all correlations presented, it is essential to state the p-values to assess the statistical significance of the results.

**Score:**

3

**Topic Fit:**

3

---

### Official Review · Reviewer_VYWE · 2025-09-15
**Review of 'Spectral Insights into Data-Oblivious Critical Layers in Large Language Models'**

**Confidence:** 4

**Review:**

This paper uses Centered Kernel Alignment (CKA) to identify critical layers in LLMs before fine-tuning. Experiments demonstrate that layers with significant representation shifts correspond to those most affected during fine-tuning, with data-independent critical layer identification.

**Key Findings**
- Strong negative correlation between CKA similarity and layer criticality across multiple models
- Critical layer rankings remain consistent across different fine-tuning datasets (data-oblivious)
- Top three principal components drive representation changes at critical layers
- Freezing critical layers reduces backdoor attack success rates significantly

**Strengths**
- Novel connection between pre-fine-tuning representation dynamics and critical layers during fine-tuning
- Data-oblivious approach with consistent rankings across different fine-tuning datasets
- Evaluation across multiple models (LLaMA-2, LLaMA-3, Phi-3)
- Spectral analysis showing top three principal components drive representation shifts
- Demonstrated practical applications in efficient fine-tuning and backdoor defense

**Weaknesses**
- Limited to decoder-only transformers
- Model scale limitation (3B-13B parameters)
- CKA may not capture all relevant representation changes

**Score:**

4

**Topic Fit:**

3

---

### Official Review · Reviewer_im1K · 2025-09-16
**Interesting directions for future work.**

**Confidence:** 3

**Review:**

This paper presents a metric based on the representations at each layer, the CKA, which the authors claims is able to predict which layers are most critical for fine tuning. They present a strong correlation between the two.

Pros:
* Writing is clear, concise, and results seem to be correct and plausible.
* Paper presents an interesting observation and opens up some avenues of future research.

Cons:
* It is not fully clear that this method is truly "Data-Oblivious" which the authors state is an important benefit of their approach. For instance on the Dolly dataset the critical layers do seem to be quite different from those on the OpenBookQA or GSM8k. A truly data-oblivious strategy might require to look directly at the model weights without reference to any dataset at all, but it is not clear this is possible.
* It isn't obvious what the key takeaway of this paper is for a practitioner. For instance, it is not clear based on Figure 5 if fine tuning only critical layers is actually helpful over the standard method of fine tuning everything.
* It would be great to see some more analysis to better understand the relationship between CKA and the critical layers beyond the strong correlation observed here. Either a theoretical or empirically validated heuristic as to what causes this behavior could prove very interesting.

**Score:**

3

**Topic Fit:**

2